# Assessment of 1863 *GRIN2A* Variants Contradicts a Role in Tumorigenesis

**DOI:** 10.3390/ijms26125558

**Published:** 2025-06-10

**Authors:** Robin-Tobias Jauss, Johannes R. Lemke, Vincent Strehlow

**Affiliations:** 1Institute of Human Genetics, University of Leipzig Medical Center, 04103 Leipzig, Germany; johannes.lemke@medizin.uni-leipzig.de (J.R.L.); vincent.strehlow@medizin.uni-leipzig.de (V.S.); 2Center for Rare Diseases, University of Leipzig Medical Center, 04103 Leipzig, Germany

**Keywords:** NMDA receptor, rare diseases, melanoma, mutation, germline, somatic

## Abstract

*GRIN2A* has previously been identified as frequently mutated in tumor samples, leading to a hypothesized involvement of *GRIN2A* in tumorigenesis. Pathogenic *GRIN2A* germline variants, on the other hand lead, to neurodevelopmental disorders, with no evidence for tumor burden. Thus, we aimed for an independent assessment of somatic and germline variation in *GRIN2A*, hypothesizing that a distinct distribution of somatic variation indicates a tumorigenic effect. All publicly available *GRIN2A* variants were obtained from ClinVar, gnomAD and Cosmic to account for germline variation in affected individuals and the general population, as well as somatic variation. Functional consequences, mutational hotspots and gene expression were assessed for each dataset to draw conclusions on the potential pathomechanisms of tumorigenesis. Pathogenic germline variants in *GRIN2A* expose a clear genotype–phenotype association and are predominantly present in functionally relevant domains, while somatic *GRIN2A* variants exhibit a uniform distribution and no high abundance in functionally relevant domains. The expression of *GRIN2A* is lower in tumor samples compared to in non-diseased tissues. Given the non-uniform distribution and domain clustering, our results suggest that specific domains of *GRIN2A* are highly intolerant towards germline variation, while a lack of somatic mutational clustering and functional relevance refutes the previously hypothesized major role of *GRIN2A* in tumorigenesis.

## 1. Introduction

Somatic variants are of particular interest to research as they harbor a much more diverse genetic variation than the germline genome. Advancements in next-generation sequencing revealed a broad spectrum of mutated genes in tumor tissue, including genes hitherto not associated with tumorigenesis. One of these genes is *GRIN2A* (HGNC: 4585), which encodes the GluN2A subunit of *N*-methyl-d-aspartate (NMDA) receptors in cerebral neurons [1]. Pathogenic germline variants in *GRIN2A* are associated with neurodevelopmental disorders and developmental and epileptic encephalopathy [2,3]. It was therefore surprising and unexpected to find that *GRIN2A* is frequently mutated in tumor samples, especially in melanomas [4,5,6,7].

The unexpected genetic variation in GRIN2A in tumor samples led to the hypothesis that *GRIN2A* might serve as a proto-oncogene [8]. Indeed, the functional characterization of identified somatic variants revealed impaired NMDA receptor functionality, leading to impaired downstream activation of known tumor suppressor genes [6]. As the analyzed somatic variants were stop-gain variants (i.e., null or nonsense variants), impaired NMDA receptor functionality could be expected. On the other hand, pathogenic germline stop-gain variants in *GRIN2A* have been associated with a less severe form of developmental delay, but not with susceptibility to tumors. Strehlow et al. [2] assessed the phenotypes of 248 individuals with *GRIN2A*-related disorders and identified only one individual with a tumor diagnosis (i.e., cerebellar glioma). Thus, from a germline perspective, the role of *GRIN2A* in tumorigenesis remains elusive.

With next-generation sequencing, the number of identified variants in databases is continuously growing for both somatic and germline variation. Publicly accessible variants can provide valuable insights into the nature of genetic variation. The consideration of all published variants in available databases is therefore a useful and unbiased method to analyze the correlation of somatic and germline variation. The previous studies on somatic variants in *GRIN2A* assessed tumor samples without assessing germline variation, while studies on pathogenic germline variants did not consider somatic variants. Our aim was therefore to evaluate variants in *GRIN2A* by integrating both genetic and somatic perspectives. We hypothesized that somatic variation in *GRIN2A* would be unique compared to germline variation, in the sense that a clustered distribution of somatic variation indicates a tumorigenic effect, which would be distinguishable from the pathomechanism in germline variation leading to a neurodevelopmental phenotype.

## 2. Results

In total, 1863 unique *GRIN2A* variants were identified in ClinVar, Cosmic and gnomAD (Appendix A). The variants were generally unique for each dataset, with only little overlap between somatic variants, germline variants and variants identified in the general population (Figure 1). Nearly two thirds of all known variants have been identified as somatic in Cosmic, and 1014 variants have been exclusively identified as somatic and are absent from the other datasets. The overlap between somatic variants and variants identified in the general population (155 variants) is higher than somatic variants identified as pathogenic germline variants (28 variants). Only two variants are present in all three datasets (c.2453C>T, p.Ala818Val and c.1510C>T, p.Arg504Trp); both variants have been identified once in the general population but are classified as (likely) pathogenic at least twice in ClinVar.

To gain insights into the distribution of these unique variants, the variant density was calculated per dataset and plotted in the position on the protein’s primary sequence (Figure 2A). As expected, pathogenic germline variants and variants identified in the general population show opposite distribution patterns: gnomAD variants are abundant in the aminoterminal domain (ATD) as well as the C-terminal domain (CTD) and depleted in the transmembrane and linker domains, where pathogenic germline variants are predominant. Somatic variants display a uniform distribution across the primary protein sequence and are not predominantly identified in any specific region of the gene. Differentiating between missense variants and null variants (e.g., frameshift and stop gain variants) does not result in a different distribution pattern, i.e., somatic missense variants are also evenly distributed on the gene (Figure 2B,C). Statistical testing for uniform variant distribution revealed a significant non-uniformity of distribution for all three datasets, but the distance (D) between observed distribution and expected uniform distribution was smallest for the somatic dataset. (Results of the Kolmogorov–Smirnov test: Cosmic: D = 0.049, *p* = 0.006; ClinVar: D = 0.358, *p* < 0.001; gnomAD: D = 0.143, *p* < 0.001. Appendix A).

To further test whether specific variant types, e.g., missense variants, are enriched in certain domains of *GRIN2A*, we visualized the variant types per domain and per dataset (Figure 3). The gnomAD dataset mostly consists of missense variants in all domains. Few other variant types were identified, mainly frameshift variants in the ATD and CTD, most of which may be subject to nonsense mediated decay. Pathogenic germline variants in ClinVar consist of an even distribution of null variants (e.g., frameshift, stop gain or canonical splice variant) in all domains and an abundance of missense variants mainly in the M-domains. Somatic variants in Cosmic harbor only a few null variants, and missense variants are evenly distributed in all domains.

As the focus of *GRIN2A*-related somatic variants was on melanoma samples, *GRIN2A* gene expression was compared for different tumor tissues including skin cutaneous melanoma tissues and paired normal tissues (Figure 4 and Appendix A). The expression of *GRIN2A* is in general higher in normal tissues compared to the respective tumor tissue. No tumor sample showed a significantly higher expression of *GRIN2A* than the matched non-disease tissue.

## 3. Discussion

Our analysis provides a comprehensive overview of somatic and germline *GRIN2A* variants, delineating their molecular consequences and highlighting the distribution of pathogenic variants. Somatic variants in *GRIN2A* are surprisingly unique and indeed represent a distinct set of variants when compared to pathogenic germline variants (Figure 1). It is therefore not surprising that these unique somatic variants have attracted research and led to the hypothesis that *GRIN2A* might play a role in tumorigenesis. However, our data indicate no particular mechanism in which *GRIN2A* might serve as a proto-oncogene, concerning neither the location nor the functional consequence of the somatic variant.

Previous studies on somatic *GRIN2A* variants analyzing missense and null variants suggested an oncogenic effect through a disruption of *GRIN2A* functionality, i.e., a loss-of-function mechanism of the NMDA receptor complex [4,5,6,7,8]. Indeed, the depletion of wildtype GRIN2A in somatic samples has been shown to increase cell proliferation in vitro, suggesting *GRIN2A* to be a tumor suppressor gene [6]. However, as germline null variants of *GRIN2A* are ubiquitously present in all tissues, a loss of NMDA receptor functionality would be expected in all tissues, not limited to potential tumor tissues of individuals carrying a pathogenic germline variant.

Pathogenic germline variants of *GRIN2A* expose a genotype–phenotype association, where null variants and missense variants in ATD and LBD S1 and S2 lead to a milder neurodevelopmental phenotype and patients with missense variants in the transmembrane or linker domain exhibit a coarse disease onset and progression [2]. This is in line with our observation of a non-uniform distribution of pathogenic germline variants and an inverse distribution of benign variants identified in gnomAD, which reveals specific regions of *GRIN2A* to be highly intolerant towards germline variation. From a germline perspective, null variants in the proximal region of the gene and variants in specific mutational hotspots can therefore be clearly identified as pathogenic and disease-causing. However, a similar association for somatic variants could not be observed, as no enrichment regarding either the location or the functional consequence can be detected for somatic variants (Figure 2 and Figure 3). A limitation of this study is the assumption that a tumorigenic effect is only relevant if these variants are present in distinct clusters or are non-uniformly distributed. Given this assumption, the uniform distribution of somatic variants across *GRIN2A* implies a random distribution of variants and a lack of mutational hotspots that are expected for the activation of tumorigenesis—or the impaired functioning of a tumor suppressor gene [9].

It is expected that oncogenes significantly contribute to tumorigenesis by altering cell proliferation and differentiation. An oncogene therefore needs to be sufficiently expressed in the respective tissue [10]. None of the analyzed tumor entities showed a significantly higher expression of *GRIN2A* when compared to normal tissue, where the expression was in some cases even higher than in the tumor sample (Figure 4 and Appendix A). In addition, except for one individual with a cerebellar glioma, tumors could not be identified among the 248 individuals with pathogenic germline *GRIN2A* variants that underwent extensive phenotyping in Strehlow et al. [2]. These findings refute the hypothesized role of *GRIN2A* as a proto-oncogene as well as having an increased tumor susceptibility.

## 4. Materials and Methods

Publicly available *GRIN2A* variants were obtained from the following three databases to account for germline variants, somatic variants and variants present in the general population: (1) ClinVar v20221103 [11], which contains variants identified in affected individuals. This dataset was filtered to exclude somatic variants, and only (likely) pathogenic germline variants were retained. (2) Cosmic v96 [12], which is a database providing variants identified in tumor tissues. The Cosmic dataset was filtered for variants that are confirmed to be somatic. (3) gnomAD v.2.1.1 [13], which is a database for variants identified in healthy controls. These three datasets were merged and variants were re-annotated with the Ensembl Variant Effect Predictor [14], after which only variants with a moderate or high impact were evaluated. Data handling and plotting was performed in R 4.1.2 and ggplot2 [15,16]. Variant distribution was plotted using Gaussian kernel density estimation (function ‘density’ in R). For the statistical testing of uniform variant distribution, a Kolmogorov–Smirnov test was applied (function ‘*ks*.test’ in R). Provided c.- and p.-codes are based on transcripts NM_001134407.3 and NP_001127879.1, respectively. Genomic positions are based on human genome assembly GRCh37. The expression of *GRIN2A* was analyzed and visualized with GEPIA [17], based on tumor expression data from The Cancer Genome Atlas (TCGA) Pan-Cancer analysis project [18] matched with non-diseased tissue expression data from the Genotype-Tissue Expression (GTEx) project [19,20]. Tumor entity abbreviations are provided in Appendix A. The code for bioinformatics analyses, data handling and generation of plots is available in a dedicated GitHub Repository: https://github.com/HUGLeipzig/GRIN2A_Tumor (accessed on 1 June 2025).

## 5. Conclusions

Our analyses integrating germline and somatic perspectives do not support the role of *GRIN2A* in tumorigenesis. While germline variants do expose a genotype–phenotype association explaining the differences in disease severity, no pattern for somatic variation could be detected. It is therefore unlikely that *GRIN2A* plays any role in tumorigenesis.

## Figures and Tables

**Figure 1 ijms-26-05558-f001:**
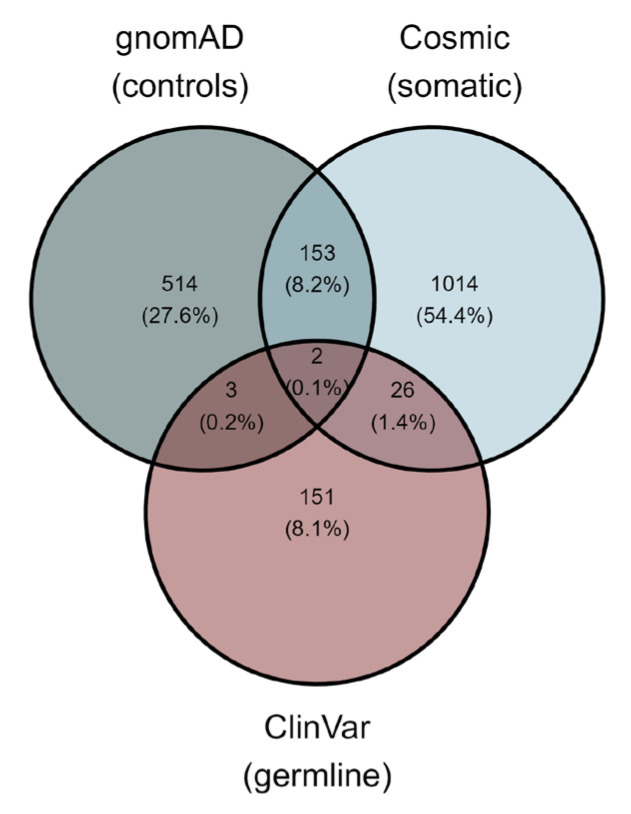
Overlap of *GRIN2A* variants identified in ClinVar, Cosmic and gnomAD, representing pathogenic germline variants, somatic variants and variants identified in the general population, respectively. Each dataset contains a distinct set of variants, with only a little overlap between the datasets.

**Figure 2 ijms-26-05558-f002:**
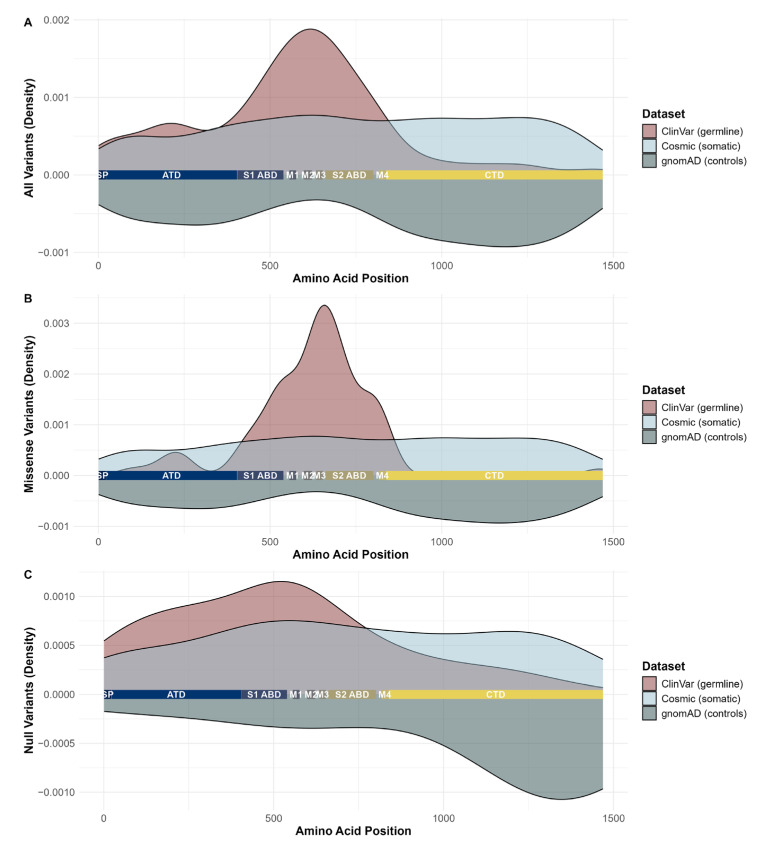
Density of germline variants identified in ClinVar, somatic variants in Cosmic and variants identified in the general population in gnomAD. (**A**): All variants; (**B**): missense variants only; (**C**): null variants only. Somatic variants are evenly distributed across the whole gene, while pathogenic germline variants are predominantly found in the ATD and linker domains. Variants identified in the general population show an inverse distribution compared to pathogenic germline variants, i.e., these variants are abundant in the CTD and depleted in the linker domains.

**Figure 3 ijms-26-05558-f003:**
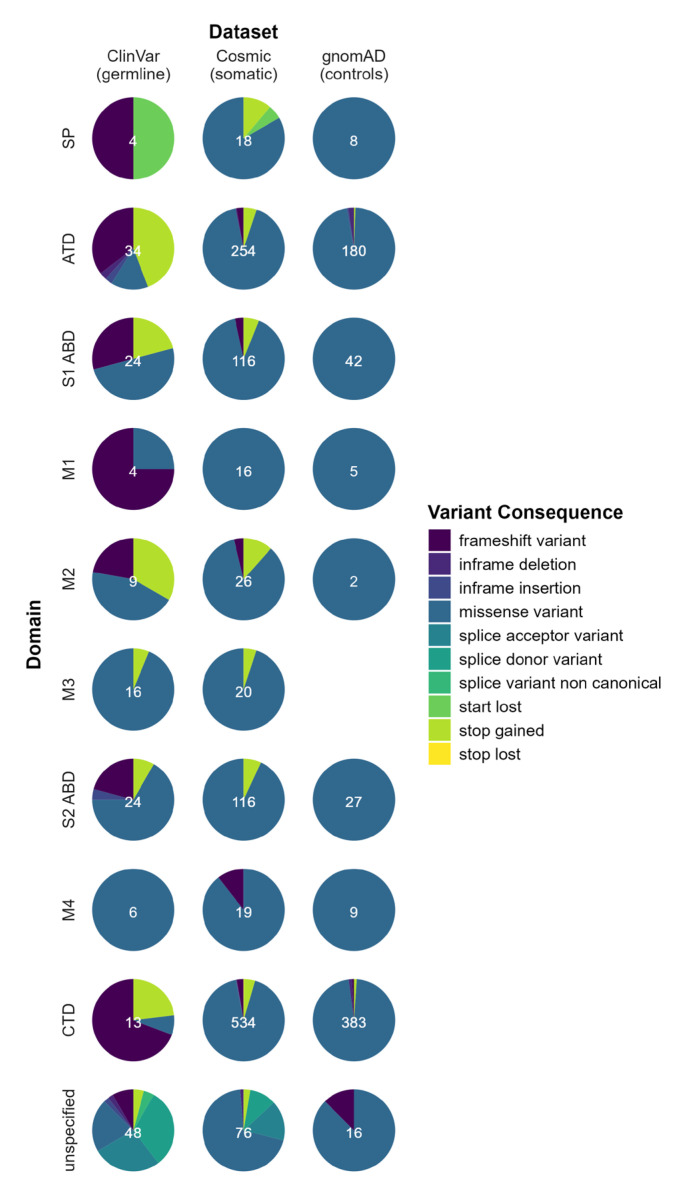
Pie charts representing the variant consequence per domain (rows) and dataset (columns). The white number in the center of each pie chart gives the total number of variants identified in this domain and dataset. No gnomAD variants were identified in the M3 domain; “unspecified” contains variants at residues not associated with a domain.

**Figure 4 ijms-26-05558-f004:**
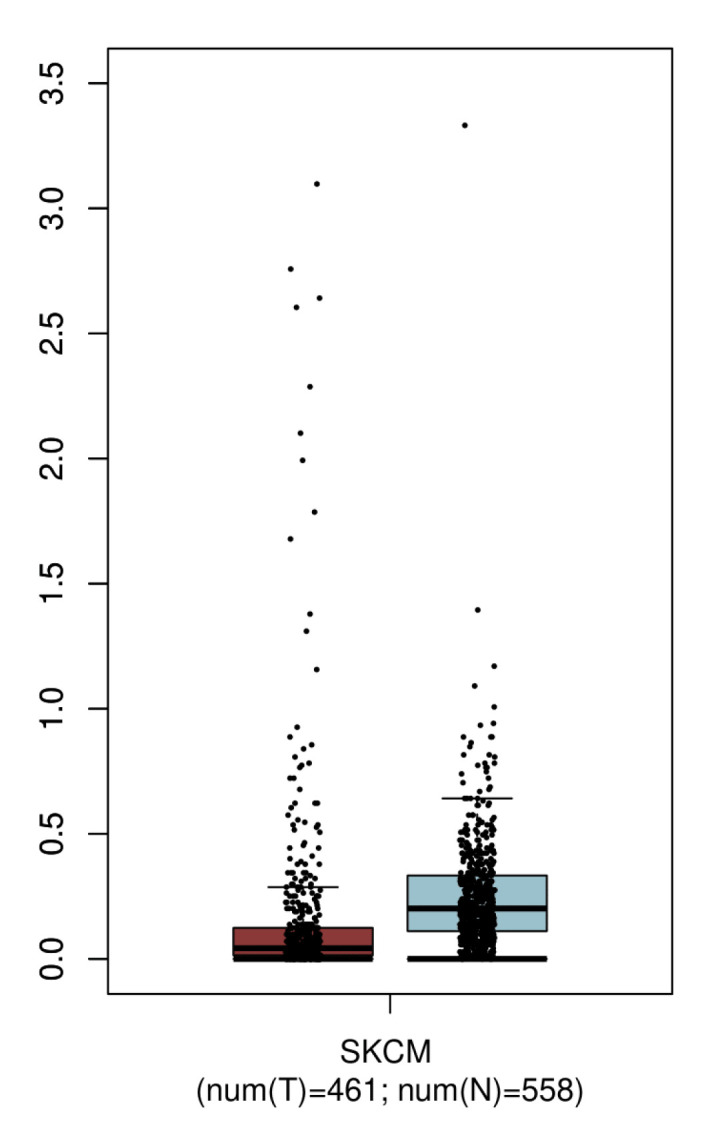
Expression of *GRIN2A* (log(TPM), *y*-axis) in skin cutaneous melanoma tissue (SKCM, red) compared to normal tissue (blue). Median expression is higher in normal tissue, although not significantly.

## Data Availability

Data generated in this study are available in the Appendix A. Further data and code used to generate the plots are available in a dedicated GitHub repository: https://github.com/HUGLeipzig/GRIN2A_Tumor (accessed on 1 June 2025).

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
