# Peer review of "Assessment of 1863 GRIN2A Variants Contradicts a Role in Tumorigenesis"

_ijms, 2025, doi:10.3390/ijms26125558_

Round 1

Reviewer 1 Report

Comments and Suggestions for Authors

I have two comments that refer to the text that was uploaded, however, I was unable to retrieve the Supplementary Data, and cannot provide a final recommendation for the paper. The paper in this present form seems unnecessarily short; I believe that including Supplementary figures in the main text, and providing a more comprehensive Discussion might solve this issue. I include the comments below that refer to the text in this stage:

  1. I would recommend that the databases are described in more detail in the Materials and Methods Section, due to the broad audience that the IJMS journal attracts. Provide more information in several sentences as to what each database collects, what it is known and used for, etc.
  2. When mentioning stop-gain variant for the first time, provide an explanation that it’s a nonsense mutation, again to account for a wider audience, since I think the “stop-gain” might be a less widely used term. Later in the text, its ok to use the one you prefer.

Reviewer 2 Report

Comments and Suggestions for Authors

This study provides a comprehensive landscape of GRIN2A variants across different biological contexts (germline, somatic, population) and highlights distinct distribution patterns that may correlate with functional and pathogenic significance. It represents a valuable resource for researchers investigating GRIN2A role in disease, especially in neurodevelopmental disorders and cancer, and sets the foundation for further functional and mechanistic studies. The study was well designed and has important findings, although it did not confirm the hypothesis. Authors integrate large-scale data from multiple reputable genomic resources, employ rigorous filtering and statistical methods, connect variant patterns to expression profiles in cancer.

The presented work contains a large number of supplementary materials, none of which are available in the peer review process:

Supplementary Materials: The following supporting information can be downloaded at: https://www.mdpi.com/article/doi/s1, Supplementary Table 1: GRIN2A variants identified in Clin- Var, Cosmic and gnomAD. Supplementary Table 2: List of tumor entity abbreviations based onTCGA nomenclature. Supplementary Figure 1: Distribution of missense variants across GRIN2A. The distribution pattern is similar when taking all variants into account (Error! Reference source not found.), as the somatic variants are evenly distributed and pathogenic germline variants are enriched in the linker domains. Supplementary Figure 2: Distribution of null variants (frameshift, stop gain etc.) across GRIN2A. Pathogenic germline variants are enriched in the ATD and ABD do- main, variants identified in the general population show an inverse distribution pattern. Again, somatic variants are evenly distributed and not enriched in a specific region of the protein product
Supplementary Figure 3: Results of Kolmogorov–Smirnov test (testing for uniform variant distribution) visualised for all three datasets Cosmic (A), ClinVar (B) and gnomAD (C). Plots show observed variant distribution across cDNA (x-axis) versus expected uniform distribution. Value “D” gives the distance between observed and expected distribution, with high values indicating higher distance from expected uniform distribution. Somatic variants in the Cosmic dataset show only little distance from uniform distribution (D = 0.049). Supplementary Figure 4: Expression of GRIN2A (log(TPM), y-axis) in skin cutaneous melanoma tissue (SKCM, red) compared to normal tissue (blue). Median expression is higher in normal tissue, although not significant. Supplementary Figure 5: Boxplots of 
GRIN2A expression (log(TPM), y-axis) in tumor samples compared to non-disease samples (x-axis). No tumor tissue shows significantly higher expression of GRIN2A compared to the corresponding normal tissue. For tumor entity abbreviations see Supplementary Table 2

 It is not possible to evaluate the results without familiarizing with these materials. Moreover, the results of the performed analyses should be made available in the form of providing a link where they can be found and viewed. Also, the entire process of bioinformatics analysis should be described in detail, which is not included in the manuscript. From minor comments: the work contains information about an incorrect reference.

Round 2

Reviewer 1 Report

Comments and Suggestions for Authors

The supplementary data is now visible, and the authors have improved the Materials and Methods section. The remaining issues are stated below.

  1. Figures 2 and 3; Supplementary Figures 1 and 2 - it would be more informative for the readers if the legends on the plot stated what the dataset represents (somatic variants, germline variants, healthy donor variants) instead of referring to which database was used for variant density plotting.

2 . Supplementary Figures 1 and 2 should be placed in the main text, and pooled into a panel with Figure 2; no need to bury the Figures in the Supplementary files.

  1. Discussion should be extended to include the comparison with studies that claim GRIN2A somatic variants are related to malignancy. Also, the results of this study seem too poorly explained to me. I think that a more detailed explanation on how variants are expected to be pathogenic is warranted.

Author Response

Comments 1: Figures 2 and 3; Supplementary Figures 1 and 2 - it would be more informative for the readers if the legends on the plot stated what the dataset represents (somatic variants, germline variants, healthy donor variants) instead of referring to which database was used for variant density plotting.

Response 1: Thank you for pointing this out. We have updated the figure legends to include a description of what the database represents, for example, we renamed "ClinVar" into "ClinVar (germline)" in all figure legends. 

Comments 2: Supplementary Figures 1 and 2 should be placed in the main text, and pooled into a panel with Figure 2; no need to bury the Figures in the Supplementary files.

Response 2: We combined Figure 2 and the Supplementary Figures into a single comprehensive Figure in the main text, thank you for your suggestion. 

Comments 3: Discussion should be extended to include the comparison with studies that claim GRIN2A somatic variants are related to malignancy. Also, the results of this study seem too poorly explained to me. I think that a more detailed explanation on how variants are expected to be pathogenic is warranted.

Response 3: We agree, and we have therefore updated the Discussion section. This section now includes a comparison with previous studies on somatic GRIN2A variants, we have also added a paragraph that explains pathogenic germline variants in more detail. We are confident that this will strengthen our Discussion and are grateful for your feedback. 

Reviewer 2 Report

Comments and Suggestions for Authors

The authors introduced the necessary corrections and information into the paper.

Author Response

Comments: The authors introduced the necessary corrections and information into the paper.
Reply: Thank you for endorsing the publication of our manuscript, we appreciate your positive consideration.